# Double Photoionization of Nitrosyl Chloride by Synchrotron Radiation in the 24–70 eV Photon Energy Range

**DOI:** 10.3390/molecules28135218

**Published:** 2023-07-05

**Authors:** Luca Schio, Michele Alagia, Robert Richter, Vitali Zhaunerchyk, Stefano Stranges, Fernando Pirani, Franco Vecchiocattivi, Marco Parriani, Stefano Falcinelli

**Affiliations:** 1IOM CNR Laboratorio TASC, 34012 Trieste, Italy; schio@iom.cnr.it (L.S.); alagia@iom.cnr.it (M.A.); stefano.stranges@uniroma1.it (S.S.); 2Sincrotrone Trieste, Area Science Park, 34149 Trieste, Italy; robert.richter@elettra.eu; 3Department of Physics, University of Gothenburg, 412 96 Gothenburg, Sweden; vitali.zhaunerchyk@physics.gu.se; 4Department of Chemistry and Drug Technology, University of Rome Sapienza, 00185 Rome, Italy; 5Department of Civil and Environmental Engineering, University of Perugia, Via G. Duranti 93, 06125 Perugia, Italy; pirani.fernando@gmail.com (F.P.); franco@vecchio.it (F.V.); marcoparriani@gmail.com (M.P.); 6Department of Chemistry, Biology and Biotechnologies, University of Perugia, Via Elce di Sotto 8, 06123 Perugia, Italy

**Keywords:** molecular dications, Coulomb explosion, electron–ion–ion coincidence technique, synchrotron radiation, potential energy surfaces, ab initio calculations

## Abstract

The behavior of nitrosyl chloride (ClNO) exposed to ionizing radiation was studied by direct probing valence-shell electrons in temporal coincidence with ions originating from the fragmentation process of the transient ClNO^2+^. Such a molecular dication was produced by double photoionization with synchrotron radiation in the 24–70 eV photon energy range. The experiment has been conducted at the Elettra Synchrotron Facility of Basovizza (Trieste, Italy) using a light beam linearly polarized with the direction of the polarization vector parallel to the ClNO molecular beam axis. ClNO molecules crossing the photon beam at right angles in the scattering region are generated by effusive expansion and randomly oriented. The threshold energy for the double ionization of ClNO (30.1 ± 0.1 eV) and six dissociation channels producing NO^+^/Cl^+^, N^+^/Cl^+^, N^+^/O^+^, O^+^/Cl^+^, ClN^+^/O^+^, NO^+^/Cl^2+^ ion pairs, with their relative abundance and threshold energies, have been measured.

## 1. Introduction

Nitrosyl chloride (ClNO) is an important reactive intermediate relevant in atmospheric chemistry as a reservoir species in the troposphere able to produce NO and Cl radicals by photodissociation, which play an important role in the atmosphere as catalytic destruction agents of ozone [1]. The ClNO molecule is easily generated in coastal urban areas by the reaction of NO_2_, a common air pollutant produced by fuel combustion, with sea salt (NaCl) airborne particles [2,3]. Its fast photolysis produces Cl atoms, whereas by hydrolysis, HONO can be formed, which is the major source of OH radicals; both species are responsible for photochemical oxidation reactions in the atmosphere.

Despite various studies already conducted concerning photo-induced processes on ClNO using photons in the ultraviolet (UV)-visible [4,5,6,7,8,9,10,11,12,13,14] and infrared [15,16,17] energy range, no data on its double photoionization by direct ejection of two valence electrons are available.

Molecular dications produced by direct photoionization are species of great interest in astrochemistry; they were found in interstellar clouds, comet tails and planetary ionospheres [18]. Moreover, dications are involved in flames and plasma chemistry and physics [18]. The importance of coincidence studies of dications produced by the direct ejection of a couple of valence-shell electrons in determining the basic chemical properties of molecules has been pointed out by the pioneering work completed by J.H.D. Eland. In the last decades, Eland’s research group has studied and characterized a large number of molecular dications of increasing complexity using double photoionization with electron–ion–ion coincidence techniques [19,20,21,22]. Despite the huge research activity already conducted and well described in various review articles [23,24,25], there is still a lack of information on dications for highly reactive species.

In this paper, we present first experimental results on the direct double photoionization of valence electrons of ClNO molecules in order to investigate and characterize the energetics and fragmentation dynamics of the nitrosyl chloride dication, ClNO^2+^ in the VUV spectral region.

The inner shell electronic structure and dissociation dynamics of nitrosyl chloride were already studied by our group at the Gas Phase Beamline of the Elettra Synchrotron Facility in Trieste (Italy) [26,27,28].

## 2. Results and Discussion

Double photoionization was studied in a previous experiment mainly dedicated to photoelectron spectroscopy (PES) and X-ray absorption spectroscopy (NEXAFS) in order to deeply characterize the electronic structure of ClNO through a combined experimental and theoretical spectroscopic investigation at the Cl 2p, Cl 2s, N 1s and O 1s edges [26]. After that, resonant Auger spectra measurements have been taken together with atomic/molecular products ion coincidence detection with resonant Auger electrons [27,28].

These previous ion–ion coincidence measurements on the ClNO dication [27,28] formed after core excitation at different pre-edge resonances have pointed out the Cl-N bond cleavage with the formation of NO^+^ ions. It has been observed that excitation of similar molecular orbitals of ClNO [26] (from the N and O atoms) results in different fragmentation patterns [27,28]. The dissociation dynamics of ClNO thus observed appear to be affected by several factors, such as the core hole state lifetime, the excitation transition moment, the vibrational motions in the excited state, the specific electronic decay channel and the topology of the potential energy surfaces (PESs) of the final state of dication reached.

The new information provided by the present study, not reported in the literature to our knowledge, is the energetic and dissociation dynamics of nitrosyl chloride dication (ClNO^2+^) produced by direct double photoionization of ClNO, as a result of a very strong electron correlation, thus avoiding the core hole formation involved in previous measurements [26,27,28]. The direct approach is the most effective type of investigation of the dications, energetics and topology of relative PESs. This study can be considered as a first attempt to investigate highly excited states of this reactive species relevant in atmospheric chemistry.

By analyzing recorded coincidence plots at each investigated photon energy (see Figure 1) using the PEPIPICO described in the Materials and Methods section, energy thresholds for several fragmentation channels produced by double photoionization of ClNO have been measured. The incident photon flux and gas pressure were monitored, and the ion yields for each fragmentation channel were divided by the total ion yield to obtain the relative cross sections as a function of the photon energy (see Figure 2).

The measured fragmentation channels are reported in the following Reactions (1)–(6):ClNO + hν → NO^+^ + Cl^+^ + 2e^−^
(1)
ClN^+^ + O^+^ + 2e^−^
(2)
O^+^ + N^+^ + Cl + 2e^−^
(3)
O^+^ + Cl^+^ + N + 2e^−^(4)
N^+^ + Cl^+^ + O + 2e^−^(5)
NO^+^ + Cl^2+^ + 3e^−^
(6)

Valuable information obtained is the threshold energy for the double ionization of ClNO (30.1 ± 0.1 eV), the other dissociation channels and their relative abundance and threshold energies. The values and uncertainties are reported in Table 1 below.

For each open dissociation channel of Equations (1)–(6), the energy threshold has been determined by using a Wannier function [29]. As we have already established in previous works [30,31,32], an empirical function is used where the threshold energy is an adjustable parameter varied until the best fit to the experimental data is achieved. Experimental errors of Table 1 reflect the scattering of experimental data. The relative abundances were estimated by adding total collected counts for each pair of product ions at all investigated photon energies.

From the relative cross sections reported in Figure 2 and from the data of Table 1, it can be appreciated that the main two-body fragmentation channel following the double photoionization of ClNO in the 24.0–70.0 eV photon energy range is the one producing the NO^+^ + Cl^+^ ion pair, with an averaged abundance of 52.7 ± 0.3%. Such a result points out differences in the dissociation dynamics of ClNO following the direct ejection of a couple of valence-shell electrons by double photoionization experiment with respect to previous resonant Auger spectra measurements by Salén et al. [27,28]. The NO^+^/Cl^+^ ion pair formation, which in our experiment is largely the most abundant fragmentation channel, in the Auger decay experiment of Salén et al. [27] is recorded as one of the less abundant channels for both resonant core-electron excitation to the LUMO and LUMO + 1 orbitals at both the N and O K edges, with a maximum NO^+^/Cl^+^ relative yield of 26.2 ± 0.1% at a photon energy of 390.0 eV, i.e., below N1 resonances. Furthermore, in their experiment, which aimed to investigate the ClNO fragmentation dynamics upon resonant core-electron excitations using the X-ray photon energy range of 390.0–534.6 eV, Salén et al. found that N^+^/O^+^, Cl^+^/N^+^ and Cl^+^/O^+^ ion pair formations are the most important ones with their relative abundances ranging between 12.5% and 30.6% [27]. On the contrary, in the present investigation, performed using photons with an energy at the threshold of the direct double photoionization, we found that such three-body dissociation channels are less abundant on average by a factor of 3.8 than the channel producing NO^+^/Cl^+^.

The relatively low abundance of the ClN^+^/O^+^ ion pair fragmentation channel (5.3 ± 0.3%, see data of Table 1) is not surprising since potential energy surface calculations performed by Yao et al. [33] at the B3LYP/6-311 G(2d,2p) and MP2/6-311 G(2d,2p) levels demonstrated that the Cl-N bond of ClNO^2+^ dication shows a very low dissociation energy. This is confirmed by results obtained in previous investigations on the double photoionization fragmentation dynamics following the core-excited states of ClNO. In particular, Salén et al. [27] found: (i) no experimental evidence for the formation of ClNO^+^ in their experiments; (ii) a low relative yield for ClN^+^/O^+^ ion pair formation which varies between 0.1% and 2.2% in the 390.0–534.6 eV X-ray photon energy range.

It has to be noted that in the investigated 24.0–70.0 eV photon energy range, we did not record any signals relating to the dissociative channels leading to the formation of O^+^/Cl^2+^, NO^+^/Cl^2+^ and Cl^+^/NO^2+^, which instead have been identified at higher photon energies by Salén et al. as minor fragmentation processes [27]. This is due to the fact that the present experiment was performed in a much lower energy range which does not allow the removal of three electrons from the ClNO molecule, with the exception of only the fragmentation channel forming the NO^+^/Cl^2+^ ion pair, which is accessible when the photon energy reaches the value of 63.5 eV (see Table 1). Moreover, from our mass spectra, we reveal the formation of Cl^2+^ atomic dications with low intensities only at photon energies greater than or equal to 45.0 eV (see doublet peak of very low intensity at m/e equal to 35/2 and 37/2 present in the coincidence spectrum at 45.0 eV in Figure 1). Despite the possible formation of N^2+^ and O^2+^, because their thresholds are also in the investigated energy range of 24–70 eV, such atomic dications have not been detected in our mass spectra. Furthermore, we are not able to give any estimation for the abundance of fragmentation channels involving Cl^2+^ atomic dications with the respective neutral products since our PEPIPICO ion imaging device (see Figure 3 and Section 3) is able to detect in coincidence only pairs of fragment ions (and not neutral species) formed in the same photoionization event.

Finally, this work demonstrates that even by performing the double photoionization of nitrosyl chloride by direct probing valence-shell electrons, there is no experimental evidence of the formation of stable or metastable ClNO^2+^ dications, in agreement with what has been observed in previous resonant Auger spectra measurements using photons in the X-ray range, and by Wang [34], who studied the dissociation of ClNO by electron impact at an energy of 200 eV. The relative abundances of the various two-body dissociation channels shown in Figure 2 reflect the dynamics of microscopic fragmentation, which follow the formation of the intermediate molecular dication ClNO^2+^ forming a pair of ionic products. In the case of our measurements, such a molecular dication is absent in the mass spectra recorded in the whole investigated photon energy range. This is a clear indication that its lifetime is shorter than 50 ns, which is the characteristic window of time of our apparatus corresponding to the time detection limit for our ion imaging detector (see Figure 3) described in Section 3. In order to be able to provide a comprehensive explanation of the trends observed for the relative cross sections as a function of the photon energy reported in Figure 2, further theoretical efforts should be made, aimed to determine the structure and energetics of the intermediate ClNO^2+^ dication. Such a computational work is in progress in our research group, together with further experimental investigations aimed to measure both kinetic energy released (KER) and angular distributions of the ion pairs emitted at the different investigated photon energies, as we have already performed in previous analogous studies [18].

## 3. Materials and Methods

The preparation and generation of the ClNO sample by our research group was carried out following Abbas et al. [35].

The energetics and fragmentation dynamics following the Coulomb explosion of the intermediate ClNO^2+^ molecular dication produced by direct double photoionization of nitrosyl chloride was studied at the “Circular Polarization” (CiPo) beamline of Elettra Synchrotron Facility (Trieste, Italy) in the 24.0–70.0 eV photon energy range. Despite the possibility of using circularly polarized synchrotron light, as provided by CiPo, in our experiments we only used linearly polarized photons with the same characteristics of the synchrotron radiation exploited in the experiments previously performed at the “Gas Phase” beamline, the target of which was to investigate the electronic structure of the inner shell and dissociation dynamics of nitrosyl chloride [26,27,28]. This was done to easily and directly compare the results obtained in valence shell double photoionization experiments versus those involving ClNO inner-shell electrons. In our experiment, the light beam is linearly polarized, and the direction of the polarization vector is aligned parallel to the molecular beam axis. The ClNO molecules crossing the photon beam at right angles in the scattering region are generated by effusive expansion and randomly oriented.

The ARPES end station (Angle-Resolved PhotoEmission Spectroscopy) apparatus for electron–ion–ion coincidence experiments, successfully employed in previous experiments, has been used [36]. The photoelectron–photoion–photoion coincidence (PEPIPICO) detector (see Figure 3) consists of a time of flight (TOF) spectrometer equipped with a position-sensitive detector especially designed to properly measure the cation photofragment momentum vectors in many body dissociation processes; it consists of a stack of three impedance-matched micro-channel plates with a multi-anode array arranged in 32 rows and 32 columns [37].

The energy selected synchrotron light beam, with a resolution of about 2.0–1.5 meV, crosses an effusive molecular beam of ClNO at right angles, and product ions are detected in coincidence with photoelectrons. A detailed description of the apparatus [37] and beamline [38] is given elsewhere.

## 4. Conclusions

In this paper, we present a study of the double photoionization of nitrosyl chloride, an important reactive species in atmospheric chemistry. In our experiment, promoted by direct ejection of two valence electrons, we were able to measure directly the first double ionization energy in the photoionization of ClNO. This study has been performed by using linearly polarized synchrotron radiation in the 24.0–70.0 eV photon energy range identifying the possible dissociation channels and measuring the threshold energy for the different ionic products’ formation and the related branching ratios. The experiment was conducted through a rather hard experimental effort in which each investigated photon energy required acquisitions of 6–8 h with an energy step of 200 meV due to the low intensities recorded in our PEPIPICO coincidence spectra.

Despite such limitations, the perspective opens for planning measurements of the kinetic-energy-released distribution of fragment ions at different photon energies with their relative angular distributions in order to investigate in greater detail the microscopic two-body dissociation mechanisms. For such a purpose, in consideration of the difficulties in producing a sufficiently large and pure quantity of the neutral molecular precursor and in performing coincidence PEPIPICO measurements with enough intensity, a dedicated future long-term beamtime will be requested. Furthermore, theoretical efforts will be performed by our group by applying a methodology already used [30,31,32]; the energy and structure of dissociation product ions could be determined, providing crucial information on the microscopic dynamics of the charge separation reactions following the double photoionization of ClNO free molecules.

## Figures and Tables

**Figure 1 molecules-28-05218-f001:**
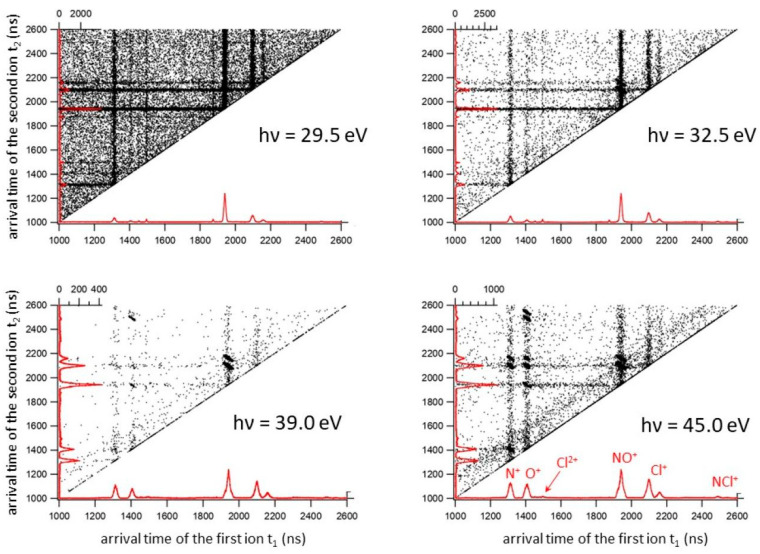
The coincidence plots recorded in the double photoionization experiment of the ClNO at four different photon energies: 29.5, 32.5, 39.0 and 45.0 eV. In these types of plots, which are typical of double photoionization experiments, the two time of flight values, t_1_ and t_2_, of two ions produced (both reported in the *x* and *y* axes in ns) in the same photoionization event define a point. In the figure, the mass spectra related to the recorded double coincidences (red color) are shown.

**Figure 2 molecules-28-05218-f002:**
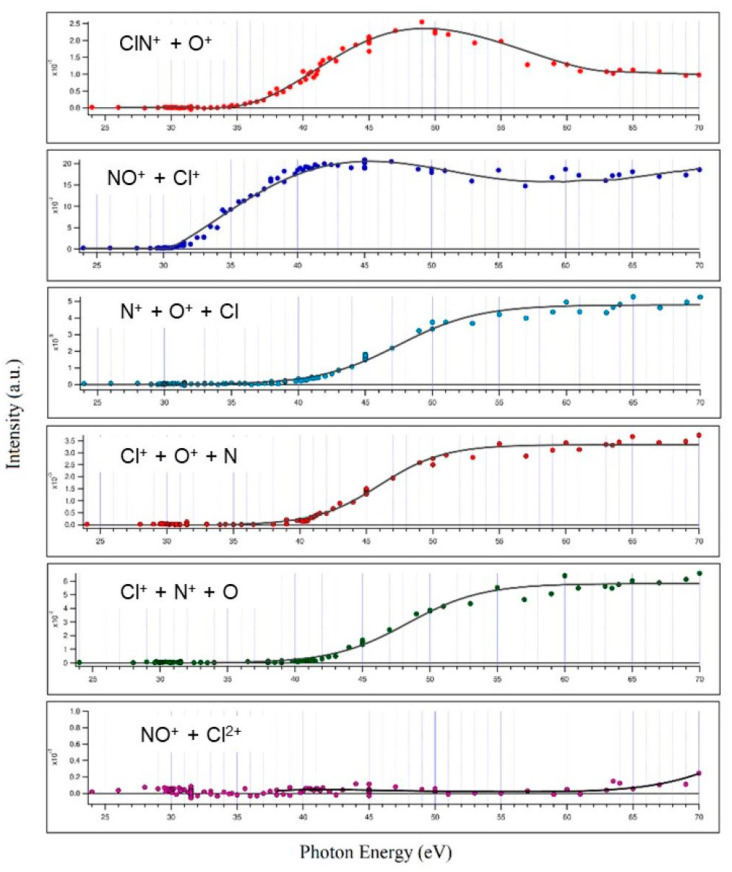
The measured relative cross sections and related threshold energies of the main fragmentation channels recorded in the double photoionization of ClNO performed in the 24.0–70.0 eV photon energy range. The channel producing NO^+^ + Cl^2+^ (Reaction (6)) is reported in the lowest panel; despite its low intensity, a careful analysis was performed allowing the threshold energy determination observed at hν ≥ 63.5 eV.

**Figure 3 molecules-28-05218-f003:**
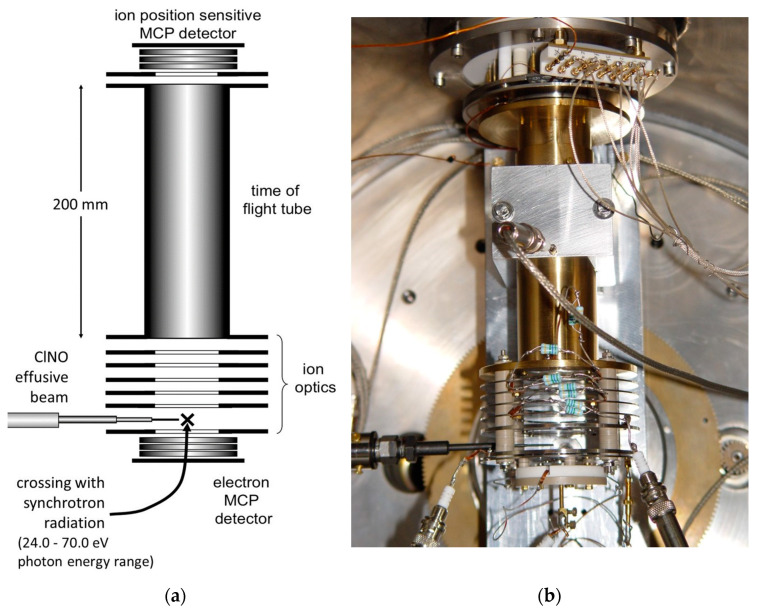
A scheme (**a**) and a picture (**b**) of the photoelectron–photoion–photoion coincidence detector used in the double photoionization experiment of ClNO (MCP stands for Micro Channel Plate detector).

**Table 1 molecules-28-05218-t001:** Products of the double ionization of nitrosyl chloride and their relative abundance, averaged over the 24.0–70.0 eV photon energy range investigated (statistical uncertainties are of 0.3%), along with the relative photon energy thresholds.

Product Ions	Average Abundance (%)	Energy Threshold (eV)
NO^+^ + Cl^+^	52.7	30.1 ± 0.1
N^+^ + Cl^+^	18.1	40.5 ± 0.1
N^+^ + O^+^	13.5	39.0 ± 0.2
O^+^ + Cl^+^	9.8	39.0 ± 0.2
ClN^+^ + O^+^	5.3	34.5 ± 0.2
NO^+^ + Cl^2+^	0.6	63.5 ± 0.3

## Data Availability

Not applicable.

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
