# Peer review of "Double Photoionization of Nitrosyl Chloride by Synchrotron Radiation in the 24–70 eV Photon Energy Range"

_molecules, 2023, doi:10.3390/molecules28135218_

Round 1
Reviewer 1 Report
This paper deals with an experimental study of the production of various fragmentation pairs of the ClNO molecule as a result of double photoionization of the valence shell. The results are compared with the case off inner-shell ionization followed by an Auger which also leads to double ionization, and it is clear that the dissociation dynamics are very different in the two cases. The paper is well-written and informative and should be published. I have two suggestions as to what else the authors might like to include.
1. Can we understand why the relative abundances are what they are from a physical point of view?
2. Can we understand the variations as a function of photon energy as depicted in Fig. 2.
It seems to me that these additions would enhance the interest of t the paper.
Author Response
Responses to the Reviewer #1 comments:
Reviewer #1 – general comments: “This paper deals with an experimental study of the production of various fragmentation pairs of the ClNO molecule as a result of double photoionization of the valence shell. The results are compared with the case off inner-shell ionization followed by an Auger which also leads to double ionization, and it is clear that the dissociation dynamics are very different in the two cases. The paper is well-written and informative and should be published. I have two suggestions as to what else the authors might like to include.”
Authors reply and made modifications: We thank the reviewer #1 for his positive comments and related useful suggestions which have been fully addressed by us (see below) since they enhance the quality and interest of our paper.
Reviewer #1 - addressed point 1: “1. Can we understand why the relative abundances are what they are from a physical point of view?”
Authors reply and made modifications: We thank the reviewer for his helpful suggestion to improve the clarity of our work, focusing on its importance from a basic physico-chemical point of view. In order to address this crucial point, we have added the following new sentences at page 5, lines 172-179:
“The relative abundances of the various two-body dissociation channels shown in Fig. 2 reflect the dynamics of microscopic fragmentation which follows the formation of the intermediate molecular dication ClNO2+ forming a pair of ionic products. In the case of our measurements, such a molecular dication is absent in the mass spectra recorded in the whole investigated photon energy range. This is a clear indication that its lifetime is unless shorter than 50 ns which is the characteristic window time of our apparatus corresponding to the time detection limit for our ion imaging detector (see Fig. 3) described in Section 3.”.
Reviewer #1 – addressed point 2: “2. Can we understand the variations as a function of photon energy as depicted in Fig. 2.”
Authors reply and made modifications: We thank the reviewer for its suggestion. To address this point, we added the following sentence at pagg. 5-6, lines 179-186, of the revised manuscript: “In order to be able to provide a comprehensive explanation of the trends observed for the relative cross sections as a function of the photon energy reported in Fig. 2, further theoretical efforts should be made, aimed to determine structure and energetics of the intermediate ClNO2+ dication. Such a computational work is in progress in our research group, together with further experimental investigations aimed to measure both kinetic energy released (KER) and angular distributions of the ion pairs emitted at the different investigated photon energies, as we have already done in previous analogous studies [18].”.

Reviewer 2 Report
The manuscript presents the experimental results for the direct double photoionization of ClNO molecule by synchrotron radiation in the 24-70 eV photon energy range, which is important for photochemical researches in astrochemistry and atmosphere. These data could also be served as a benchmark for subsequent theoretical studies. Therefore, I would recommend the publication of this communication if the following issues are appropriately addressed:
(1) In Sec.3, the authors mentioned the “Circular Polarization” beamline were used for the measurement, while in Sec. 4 the word became “linearly polarized synchrotron radiation”, which make some confusion. If the linearly polarized photon were used in the experiment, does the molecule can still be considered as a random orientation one? I think the author should state clearly the information of photon polarization and molecule orientation in the manuscript, including the “Abstract” and “Results and Discussion” parts.
(2) As a completeness, it should be better if the authors could give an estimation for the abundance of fragmentation channels involving the atomic dications with the respective neutral products, such as Cl^2+, N^2+ and O^2+, because their thresholds are also in the energy range of 24-70 eV.
(3) In line 29, “costal urban” should read as “coastal urban”.
Author Response
Responses to the Reviewer #2 comments:
Reviewer #2 – general comment: “The manuscript presents the experimental results for the direct double photoionization of ClNO molecule by synchrotron radiation in the 24-70 eV photon energy range, which is important for photochemical researches in astrochemistry and atmosphere. These data could also be served as a benchmark for subsequent theoretical studies. Therefore, I would recommend the publication of this communication if the following issues are appropriately addressed:…”
Authors reply and made modifications: We thank the reviewer #2 for his very positive comments and helpful suggestions aimed to improve the quality of our manuscript. We revised the manuscript addressing all his comments and suggestions as it follows.
Reviewer #2 – addressed point 1: “(1) In Sec.3, the authors mentioned the “Circular Polarization” beamline were used for the measurement, while in Sec. 4 the word became “linearly polarized synchrotron radiation”, which make some confusion. If the linearly polarized photon were used in the experiment, does the molecule can still be considered as a random orientation one? I think the author should state clearly the information of photon polarization and molecule orientation in the manuscript, including the “Abstract” and “Results and Discussion” parts.”.
Authors reply and made modifications: We agree and thank the reviewer for his useful suggestion. In order to address this point, avoiding any confusion, we have added at pag. 6, lines 193-203, the following sentences: “Despite the possibility of using circularly polarized synchrotron light, as provided by CiPo, in our experiments we only used linearly polarized photons with the same characteristics of the synchrotron radiation exploited in the experiments previously performed at the "Gas Phase" beamline, whose target was to investigate the electronic structure of the inner shell and dissociation dynamics of nitrosyl chloride [26-28]. This was done to easily and directly compare the results obtained in valence shell double photoionization experiments versus those involving ClNO inner shell electrons. In our experiment the light beam is linearly polarized and the direction of the polarization vector is aligned parallel to the molecular beam axis. The ClNO molecules crossing the photon beam at right angles in the scattering region are generated by effusive expansion and randomly oriented.”;
Furthermore, according to the suggestion of the reviewer#2, we have added the following sentence in the Abstract session, at page 1, lines 18-22: “The experiment has been done at the Elettra Synchrotron Facility of Basovizza (Trieste, Italy) using a light beam linearly polarized with the direction of the polarization vector parallel to the ClNO molecular beam axis. ClNO molecules crossing the photon beam at right angles in the scattering region are generated by effusive expansion and randomly oriented.”.
Reviewer #2 – addressed point 2: “(2) As a completeness, it should be better if the authors could give an estimation for the abundance of fragmentation channels involving the atomic dications with the respective neutral products, such as Cl2+, N2+ and O2+, because their thresholds are also in the energy range of 24-70 eV.”
Authors reply and made modifications: We thank the reviewer for the useful suggestions aimed at better clarifying and complete the discussion of our data presented. In order to address this point, we added the following sentences at page 5, lines 157-166, of the revised manuscript version: “Moreover, from our mass spectra, we reveal the formation of Cl2+ atomic dications with low intensities only at photon energies greater than or equal to 45.0 eV (see doublet peak of very low intensity at m/e equal to 35/2 and 37/2 present in the coincidence spectrum at 45.0 eV in Fig. 1). Despite the possible formation of N2+ and O2+, because their thresholds are also in the investigated energy range of 24-70 eV, such atomic dications have not been detected in our mass spectra. Furthermore, we are not able to give any estimation for the abundance of fragmentation channels involving Cl2+ atomic dications with the respective neutral products since our PEPIPICO ion imaging device (see Fig. 3 and Section 3) is able to detect in coincidence only pairs of fragment ions (and not neutral species) formed in the same photoionization event.”.
Reviewer #2 – addressed point 3: “(3) In line 29, “costal urban” should read as “coastal urban”.
Authors reply and made modifications: We agree with the reviewer and apologize for the error in the text. We have corrected the sentence as it can be seen in the revised manuscript version. Many thanks.
